# Thiol-Substituted Poly(2-oxazoline)s with Photolabile Protecting Groups—Tandem Network Formation by Light

**DOI:** 10.3390/polym12081767

**Published:** 2020-08-07

**Authors:** Niklas Jung, Fiona Diehl, Ulrich Jonas

**Affiliations:** Macromolecular Chemistry, Department Chemistry-Biology, University of Siegen, Adolf-Reichwein-Strasse 2, 57076 Siegen, Germany; Niklas.Jung@uni-siegen.de (N.J.); Fiona.Diehl@uni-siegen.de (F.D.)

**Keywords:** in situ-forming gels, polymer network, chemical bonding, thiol-ene click, poly(2-oxazoline)s, photoremoveable protecting group (PPG)

## Abstract

Herein, we present a novel polymer architecture based on poly(2-oxazoline)s bearing protected thiol functionalities, which can be selectively liberated by irradiation with UV light. Whereas free thiol groups can suffer from oxidation or other spontaneous reactions that degrade polymer performance, this strategy with masked thiol groups offers the possibility of photodeprotection on demand with spatio-temporal control while maintaining polymer integrity. Here, we exploit this potential for a tandem network formation upon irradiation with UV light by thiol deprotection and concurrent crosslinking via thiol-ene coupling. The synthesis of the novel oxazoline monomer 2-{2-[(2-nitrobenzyl)thio]ethyl}-4,5-dihydrooxazole (NbMEtOxa) carrying 2-nitrobenzyl-shielded thiol groups and its cationic ring-opening copolymerization at varying ratios with 2-ethyl-2-oxazoline (EtOxa) is described. The tandem network formation was exemplarily demonstrated with the photoinitator 2-hydroxy-2-methylpropiophenone (HMPP) and pentaerythritol tetraacrylate (PETA), a commercially available, tetrafunctional vinyl crosslinker. The key findings of the conducted experiments indicate that a ratio of ~10% NbMEtOxa repeat units in the polymer backbone is sufficient for network formation and in-situ gelation in *N*,*N*-dimethylformamide.

## 1. Introduction

Poly(2-oxazoline)s are a class of pseudo-polyamides with a polyethyleneimine backbone and pendant amide substituents [1]. Interesting features of this polymer class encompass antifouling [2], biocompatibility [3,4,5], or therapeutic [1,6] properties. Consequently, in recent years, a large body of research was addressed to this domain [7,8,9]. For many applications of these poly(2-oxazoline)s, e.g., antifouling surface coatings, crosslinked networks are required [10]. In the literature, various crosslinking mechanisms are described to yield such network architectures, specifically as water-swollen hydrogels. Pioneering work was reported by Seagusa and coworkers at the end of the last century [11,12,13,14]. Several years later, different research groups regained interest in poly(2-oxazoline)s as potential precursors for hydrogels [15,16]. A particular elegant strategy to crosslink poly(2-oxazoline)s harnesses the power of the thiol-ene click reaction, as demonstrated by many groups [10,17,18,19,20,21,22,23,24]. There, vinyl-functionalized poly(2-oxazoline)s as network backbone and multi-functional thiol derivatives as small crosslinker units were utilized [22].

To our knowledge, so far no gel has been reported that has been formed from a thiol-functionalized poly(2-oxazoline) and a crosslinker with unsaturated carbon bonds. One advantage of this novel concept concerns the increased stability (e.g., against oxidation) and accessibility of numerous vinyl- or alkyne-functionalized crosslinker derivatives opposed to their corresponding thiol analogues. Furthermore, the complementary thiol-substituted poly(2-oxazoline)s represent by themselves an interesting class of functional polymers:a.Targeted functionalization of poly(2-oxazoline)s may be conveniently performed utilizing thiol groups;b.The biological activity of free thiol groups (e.g., after conversion to asymmetric disulfides) as biologically cleavable linker, which has been proposed in advanced transfection methodologies [25];c.The thiol group can serve as anchor group for immobilization of poly(2-oxazoline)s on metal surfaces like Au and Ag [26,27];d.In self-healing materials, the thiol moieties can undergo so-called thiol-disulfide exchange reactions, which is an interesting motive for repair processes [28,29].

Despite these advantages of free thiol groups, such SH–modified poly(2-oxazoline) derivatives may be affected by the following complications:a.2-Oxazoline monomers bearing free thiol groups are not applicable in cationic ring opening polymerization (CROP), since the thiol is a nucleophile that acts as termination agent for the cationic active centers [30];b.Further, unprotected thiol groups are susceptible to oxidation to disulfides (e.g., with oxygen from air), which renders them inactive in thiol-ene click reactions and consequently may lead in poly(2-oxazoline)s to uncontrolled crosslinking, making storage and handling under ambient conditions difficult;c.Free thiol-groups in poly(2-oxazoline)s could also interfere in biological applications with sulfur-containing biomolecules (in particular in living organisms) by nucleophilic or disulfide exchange reactions [10].

Furthermore, thiol moieties may adversely transform other reactive groups within the same polymer backbone, e.g., reduction of azide moieties [31]. Consequently, protection of these free thiol moieties along the polymer backbone is necessary. For many side chain functionalities in poly(2-oxazoline)s, e.g., aldehyde [32], amino [33], and carboxylic acid groups [34], a large number of protecting groups have been reported in the literature. So far, for mercapto-modified poly(2-oxazoline)s, only the 4-methoxybenzyl substituent has been discussed as protecting group [35]. Its cleavage requires quite harsh conditions with trifluoroacetic acid and anisole under elevated temperatures over extended periods of time [35], which may cause damage to sensitive moieties in the polymer.

An alternative approach employs photoremovable protecting groups (PPGs), which have been reported for various functionalities [36,37,38,39,40,41,42,43,44]. One of the early PPGs, which found broad application, is based on the 2-nitrobenzyl motif, and till the 1980s about six main categories of different PPG classes were described [41]. Until 2013, the spectrum of PPG systems was significantly enhanced, as elaborated by the exhaustive review of Klán et al. [36]. Some current research directions that are worthwhile to point out are as follows: concurrent implementation of complementary PPGs at solid interfaces, which allow selective deprotection in a lithographic manner [42]; long-wavelength-sensitive PPGs for biological and biomedical in vivo applications [43]; and two-photon-sensitive PPGs, which combine the advantages of longer wavelengths for deprotection with high spatial resolution [44]. Generally, PPGs provide the advantage of using light as sole means for cleavage under mild conditions, allowing for good spatio-temporal control and obviating the need for chemical reagents in the deprotection reaction [36,40].

In the present publication, we focus on the well-known photolabile 2-nitrobenzyl unit as PPG for novel thiol-modified poly(2-oxazoline)s. The efficient synthesis of an oxazoline monomer carrying a 2-nitrobenzyl-shielded thiol group and its copolymerization with 2-ethyl-2-oxazoline (EtOxa) is reported. As a proof of principle, the tandem reaction cascade depicted in Figure 1 was investigated with the synthesized polymer system by preliminary gelation experiments. This cascade comprises photodeprotection and simultaneous thiol-ene click coupling to result in network formation.

## 2. Materials and Methods

Experimental details about the synthetic procedures and characterization methods are provided in the Appendix A.

## 3. Results and Discussion

### 3.1. Monomer Synthesis

The novel monomer 2-{2-[(2-nitrobenzyl) thio]ethyl}-4,5-dihydrooxazole (**4**), abbreviated NbMEtOxa, was prepared in four steps partially based on concepts from the literature (see Scheme 1) [45]. For the synthesis of 3-[(2-nitrobenzyl)thio] propanoic acid (**1**), the thiol group of mercaptopropionic acid was protected with *o*-nitrobenzyl bromide, according to reaction procedures published by Hensarling et al. and Pauloehrl et al. [46,47]. In the subsequent step, the carboxyl group was converted to the *N*-hydroxysuccinimide ester **2** and directly coupled with 2-chloroethylamine to yield amide **3**. Ring closure to the 2-oxazoline **4** was accomplished with potassium carbonate at elevated temperatures in acetonitrile. The overall yield of the 4-step synthesis route amounted to 47%, and corresponding ^1^H and ^13^C NMR spectra (see ESI, Appendix A) confirmed the presence of the title compound **4**.

### 3.2. Polymer Synthesis 

Copolymerization of NbMEtOxa with other 2-oxazolines allows one to incorporate properties of the respective comonomers into a single polymer architecture. The copolymer composition can be conveniently controlled by this approach via the comonomer mixing ratio in the reaction feed. In particular, for copolymers based on 2-alkyl-2-oxazolines, the thermoresponsivness in aqueous solution can be controlled by the types of alkyl groups and their combinations [8]. In our study, we employed commercially available EtOxa as a comonomer, which has been extensively used by many groups. After deprotection of the NbMEtOxa repeat unit, the free thiol groups were exploited for crosslinking of the copolymer, with the crosslinking density being tailored by the comonomer ratio of NbMEtOxa and EtOxa. Several polymerization attempts of the novel NbMEtOxa indicated problems with the common initiator methyl trifluoromethanesulfonate (MeOTf), as no reasonable amounts of polymer could be obtained. A potential impediment may be S-methylation by MeOTf, analogous to issues reported by Cesana and coworkers for the copolymerization of amino-functionalized oxazolines [33]. Various literature reports suggest a strategy with a preinitiator species for functionalized 2-oxazolines to circumvent such side reactions. There, a preceding reaction of an appropriate 2-alkyl-2-oxazoline and methylation agent (e.g., MeOTf or methyl tosylate) generates an initiator species for the subsequent polymerization of the problematic oxazoline monomers [32,33,35,48]. Based on these reports, we found that oligo(2-ethyl-2-oxazoline)-2-ethyl-2-oxazolinium trifluoromethanesulfonate (EtOxaOTf, formed from EtOxa and MeOTf) allowed successful copolymerization of EtOxa and NbMEtOxa. The copolymers with different NbMEtOxa contents (**P1%**, **P2.5%**, **P5%**, and **P10%**) were characterized by ^1^H NMR spectroscopy, gel permeation chromatography (GPC), and UV/Vis spectroscopy. Integration of both types of monomers into the same polymer backbone is corroborated by their respective proton signals in the ^1^H NMR spectrum, as exemplarily shown for **P10%** in Figure 2A. The relative ratio of 8.5 mol% for the NbMEtOxa repeat units in the polymer could be determined by comparing the integral of signal **a** (*δ* = 3.02 ppm) with that of signals **f** (*δ* = 7.41–7.93 ppm), which is in good agreement with the monomer feed of 10 mol%. The molar mass at peak maximum (*M*_p_) of the copolymers, as determined by GPC (see ESI, Appendix A), matches the low NbMEtOxa content with the target molar mass of 15 kg mol^−1^, but decreases with higher comonomer content (*M*_p_ for P1%: 16.1 kg mol^−1^; P2.5%: 13.7 kg mol^−1^; P5%: 8.0 kg mol^−1^; P10%: 5.6 kg mol^−1^). For an ideal Gaussian curve, ${\bar{M}}_{n}$ corresponds to *M*_p_, but due to the broader molar mass distributions and the presence of several polymer species ${\bar{M}}_{n}$ is not well defined and thus we provide *M*_p_ for the maximum of the GPC trace. Molar mass distributions (*Ð* = 1.3–3.2) were broader than expected for CROP of 2-oxazolines (*Ð* typically around 1.2). These GPC traces show several local maxima that indicate the presence of multiple species. In addition, the UV detector signal deviates significantly from the RI detector signal. The aromatic protecting group of NbMEtOxa is the only functionality that can contribute to a UV absorption at the specific detector wavelength (*λ* = 280 nm), while the EtOxa repeat unit possesses no such UV absorption. A deviation between the shapes of the UV and RI detector signal traces reflects a compositional drift for different molar masses. In order to quantify the absolute number of incorporated comonomer units, the corresponding response factors of each repeat unit type and both detectors must be known [49]. The response factor is defined by the specific detector sensitivity for a given type of repeat unit. For the copolymers discussed in this paper, no response factors can be determined, either for the EtOxa repeat unit, as it does not show any UV absorption, or for the NbMEtOxa repeat unit, as it does not form defined homopolymers. Nevertheless, a qualitative analysis of the compositional drift is possible by the following procedure to generate a UV-RI difference plot: (a) first, the UV and RI traces are normalized by dividing all signal intensity values with the one at the respective trace maxima; (b) then, the normalized RI trace is subtracted from the normalized UV trace. Even though this normalization procedure does remove information on the absolute number of repeat units in the polymer backbone and thus prevents quantitative analysis, this internal calibration yet allows qualitative comparison of the compositional drift between different samples.

For the example of a perfectly random copolymer with fixed comonomer ratio, the proportionality between the UV and RI detector traces is independent of the degree of polymerization, as the overall composition is identical for every chain length. As consequence, the UV-RI difference plot would result in a straight line at zero value. In the case of a polymer composed of non-UV-active monomers, this procedure would yield the inverse of the normalized RI trace with -1 at minimum. In contrast, if the copolymer composition of UV-absorbing and non-UV-active comonomers varies for different chain lengths, non-zero values are expected in such a UV-RI difference plot. Positive values in the UV-RI difference plot indicate polymer chains containing higher amounts of UV-absorbing units, while negative values represent the same but with higher content of non-UV-active comonomer in reference to the chain composition at the elution volume, where the normalized intensity of the UV trace equals that of the RI trace (intersection of both curves).

In our polymers, the significant deviations from the zero-line (abscissa) imply that the ratio between the UV-absorbing NbMEtOxa-related repeat units and the non-UV-active EtOxa repeat units is not uniform for all polymer chain lengths (see ESI, Appendix A). In essence, the UV-RI difference plots for all polymer samples (**P1%**, **P2.5%**, **P5%**, and **P10%**) show a positive peak at low molar masses (short chain lengths at high elution volumes) and a negative deflection in the intermediate region, while at high molar masses (long chains at low elution volumes) the amount of UV-absorber seems to increase again. The data suggest the coexistence of three different polymer architectures:a.Low molar mass polymers (fraction 1) with high NbMEtOxa content that can be removed by deprotection (see discussion on photodeprodection further below);b.Polymer chains (fraction 2) consisting of high EtOxa and low NbMEtOxa content; c.High molar mass species (fraction 3) with UV-active groups.

The low molar mass polymers (fraction 1) with high content of UV-active species may be explained by a tendency for early chain termination of NbMEtOxa. This assumption is supported by the fact that the homopolymerization of NbMEtOxa solely led to oligomers even by employing the above presented two-step initiation strategy (see ESI, Appendix A). In contrast, the EtOxa monomer supports extended chain growth leading to longer polymer chains with lower NbMEtOxa content (fraction 2). Furthermore, the *o*-nitrosobenzaldehyde PPGs from the NbMEtOxa repeat units in the polymer backbone may undergo side-reactions that lead to branched polymer species with high molar masses (fraction 3) and a high content of UV-active groups.

### 3.3. Photodeprotection 

Deprotection of the copolymers was performed in acetonitrile solution (*c* = 0.25 mg ml^−1^) by irradiation with UV light at a wavelength of *λ* = 365 nm and radiant exposure of up to *H_e_* = 24 J cm^−2^ (exposure time up to 2 h). Under these conditions, photoinduced cleavage of the *o*-nitrobenzyl PPG follows the mechanism of a Norrish type II reaction, forming 2-nitrosobenzaldehyde as side product (chemical reaction depicted in Figure 2) [36,39,40]. Successful fragmentation is clearly indicated by the decrease of the *o*-nitrobenzylic ^1^H NMR signal intensities at *δ* = 4.09 ppm (signal **e**) and *δ* = 7.41–7.93 ppm (signal **f**) in Figure 2A,B. The deprotection efficiency was determined by comparison of the signal integrals at *δ* = 4.09 ppm (signals **e** and **e’**) before and after irradiation, amounting to 90% after 2 h irradiation. Similar values are reported in the literature for carboxylates, which were protected with the same PPG [40]. Delaittre and coworkers reported quantitative deprotection of *o*-nitrobenzyl-masked thiol groups in polyacrylamide RAFT copolymers after 2 h irradiation time and workup [50]. In our case, extended irradiation times beyond 2 h showed no further increase in deprotection yield, potentially due to side reactions during CROP of the incorporated NbMEtOxa units, as discussed above for the high molar mass species (fraction 3). After deprotection, the UV-RI difference curves show a substantial loss of UV-active components for the low molar mass region (see ESI, Appendix A). The GPC traces indicate a high molar mass fraction with residual UV absorbance that result from persistent UV-active moieties that cannot be removed by light (see ESI, Appendix A). These moieties may be a consequence from NbMEtOxa side-reactions that yield branched polymers, which become insensitive to UV deprotection. Furthermore, additional high molar mass species are observed in the GPC traces of the deprotected samples, which may result from oxidative coupling of the liberated thiol groups.

Comparison of the UV/Vis spectra (Figure 2C), recorded before (solid lines) and after 2 h irradiation (dashed lines), showed an absorbance increase in the range of *λ* = 300–350 nm, proportional to the concentration of the liberated *o*-nitrosobenzaldehyde with two local absorption maxima at 285 nm and 310 nm [50,51]. In Figure 2D, the time-dependent change of absorbance during deprotection is exemplarily shown for **P10%** at a concentration of *c* = 0.25 mg ml^−1^. With progressing time, the *o*-nitrosobenzaldehyde absorbance in the range of *λ* = 270–350 nm steadily increases, as indicated by the red arrows. Interestingly, two additional temporary absorbance maxima around *λ* = 265 and 350 nm appear during the first 20 min of irradiation and decrease again (indicated by the bent yellow arrows), which may be attributed to reactive intermediates [40,51,52,53]. The kinetic measurements in Figure 2E were performed using four different concentrations of **P10%**, as indicated in the figure legend, by following the absorbance of the *o*-nitrosobenzaldehyde maximum at *λ* = 310 nm. Independent of the polymer concentration, the UV/Vis measurements show no further change in absorbance beyond 80 min of irradiation, indicating maximal deprotection of the thiol moieties. The small variations (flattening) in the kinetic curves for the different polymer concentrations can be attributed to an internal filter effect caused by the released *o*-nitrosobenzaldehyde that absorbs the incident light [36,40] and therefore decreases the amount of photons triggering the photoelimination.

### 3.4. Tandem Network Formation 

The free thiol groups along the polymer backbone provided by photodeprotection are a prerequisite for network formation by reaction with a separate crosslinker unit. Our proposed tandem reaction cascade employs photoactivated thiol-ene click coupling between such liberated thiol groups and vinyl moieties in the crosslinker. Specifically, pentaerythritol tetraacrylate (PETA) was used as tetrafunctional crosslinker and 2-hydroxy-2-methylpropiophenone (HMPP) as photoinitiator for radical activation. The reaction conditions reported by Ooi and coworkers were adapted to the increased molar ratio of thiol groups along the polymer backbone [54]. Crosslinking tests (Figure 3) were performed with 15 wt% polymer solutions of **P1%**, **P2.5%**, **P5%**, and **P10%**, in DMF, using UV light at a wavelength of *λ* = 365 nm and radiant exposure of up to *H_e_* = 24 J cm^−^² (exposure time up to 2 h, each experiment with five replicates). Successful network formation is demonstrated in Figure 3 with a firm gel that retains its oblique shape when inverting the vial containing the mixture that was irradiated in a tilted position. Even though the initial photoexcited state during deprotection of the *o*-nitrobenzyl PPG may involve radical species [39], it was found that addition of the photoinitiator HMPP was crucial for gel formation via the thiol-ene click process. As no gel formation was observed in the absence of the photoinitiator, spontaneous thiol-Michael addition was excluded under these conditions. Neither did irradiation of a DMF solution of photoinitiator HMPP and the tetrafunctional acrylate crosslinker PETA (ESI, Appendix A, vial 2) nor irradiation of a mixture of HMPP and **P10%** (ESI, Appendix A, vial 1) alone lead to gel formation. In the presence of photoinitiator and crosslinker, successful network formation was achieved for **P10%** with a higher NbMEtOxa content after an irradiation time of 1 h by the formation of a non-flowing gel. In contrast, for copolymers containing lower amounts of NbMEtOxa (**P1%**, **P2.5%**, **P5%**) no gel formation was observed, even after doubling either the irradiation time, the photoinitiator concentration, or the polymer mass concentration. In order to preserve the positive intrinsic properties of poly(EtOxa), the NbMEtOxa content was limited to 10%, as this amount was already sufficient for crosslinking.

## 4. Conclusions

In summary, we could demonstrate a novel tandem process for the photoinduced formation of a poly(oxazoline) network involving concurrent photodeprotection of polymer-associated thiol units and their thiol-ene click coupling to a vinyl crosslinker agent. For this purpose, a synthesis route for the novel monomer NbMEtOxa with overall yield of 47% was developed. Copolymerization with EtOxa led to polymers with masked thiol functions, for which nearly quantitative photodeprotection was achieved within 2 h at *λ* = 365 nm (*H_e_* = 24 J cm^−2^). Successful network formation via the tandem deprotection-crosslinking route requires a minimum ratio of NbMEtOxa repeat units (in the investigated system around 10%) in the macromolecular structure.

The unique feature of this strategy lies in the concurrent activation of functional groups and induction of a crosslinking reaction both triggered by light with high spatio-temporal control. This approach circumvents potential side reactions of free thiol groups that affect polymer stability and allow storage and convenient handling of a thiol-containing polymer.

Further studies will be directed to the improvement of the gelation efficiency at lower thiol group content (other types of crosslinkers, initiators, and solvents); extension of the tandem network formation concept to responsive polymer systems; and the exploration of potential applications in materials, biomedical, and surface sciences.

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
