# Peer review of "Thiol-Substituted Poly(2-oxazoline)s with Photolabile Protecting Groups—Tandem Network Formation by Light"

_polymers, 2020, doi:10.3390/polym12081767_

Round 1

Reviewer 1 Report

Journal: Polymers

Manuscript ID: polymers-890427

Title: Thiol-Substituted Poly(2-Oxazoline)s with Photolabile Protecting Groups - Tandem Network Formation by Light

This manuscript describes a novel polymer architecture based on poly(2-oxazoline)s bearing protected thiol functionalities, and studies the tandem network formation upon irradiation with UV light by thiol deprotection and concurrent crosslinking via thiol-ene coupling. In general, this work is well conducted and organized. Therefore, I recommend the paper for publication in Polymers after revision. My comments are as follows.

Comments:

  1. Keywords are used to reflect the main content of the article, and play an important role in document retrieval. The authorschose too many keywords. And “CROP” and “synthesis and functionalization” are inappropriate as keyword
  2. In Page 2, the article lacks current research progresson the application of photoremovable protecting groups. And what are the advantages of this research compared with other researches?
  3. In “Polymer synthesis”section, the authors describe that “the number averaged molar masses (`Mn) of the copolymers are congruent with the target molar mass of 15 kg mol-1”. What is the number averaged molar masses (`Mn) of the copolymers determined by GPC?
  4. Some figures are not clear enough to be read and the quality should be improved.In Figure S15, I suggest that the GPC curves of the copolymer samples (p1% , p2.5% , p5% , p10%) are placed on a graph to better illustrate the variation trend of molar mass with the increase of NbMEtOxa content.
  5. When the copolymer contains a small amount of NbMEtOxa (P1%, P2.5%, P5%), the gel does not form.For the polymers with lower thiol group content, how to improve their gel efficiency?
  6. The abbreviation that appears for the first time in this article needs to be written inits full name. For example, “CROP” in Keywords.
  7. There are someformatting errors and grammatical errors in this paper. For example, in Page 1 Line 24, “(PPG), polymer network” should be corrected as “(PPG); polymer network”. In “Supporting Information” section, “20 µl” should be corrected as “20 µL”. In Page 8 Line 16, “Since in the absence of the photoinitiator no gel formation was observed, spontaneous thiol-Michael addition under these conditions is” needs to be modified. There are other similar mistakes, which aren’t listed here, please revise after checking carefully.

Author Response

Reply to the Comments of Editor and Reviewers

Journal:            Polymers

Manuscript ID:  Polymers-890427

Title:                 Thiol-Substituted Poly(2-Oxazoline)s with Photolabile Protecting Groups - Tandem Network Formation by Light

We would like to thank the editor and both reviewers for their very useful and constructive remarks, which give us the opportunity to revise and to improve our manuscript. We first quote a comment from the editor’s and reviewers’ reports, then provide our specific reply, followed by details of the relevant revisions made to the manuscript (manuscript citations written in italics).

Reviewer #1:

Reviewer #1: “This manuscript describes a novel polymer architecture based on poly(2‑oxazoline)s bearing protected thiol functionalities, and studies the tandem network formation upon irradiation with UV light by thiol deprotection and concurrent crosslinking via thiol-ene coupling. In general, this work is well conducted and organized. Therefore, I recommend the paper for publication in Polymers after revision. My comments are as follows.”

Our Answer: “Thank you very much for your kind evaluation and please find our responses and revisions to all your comments below.”

Reviewer #1: “Keywords are used to reflect the main content of the article, and play an important role in document retrieval. The authors chose too many keywords. And “CROP” and “synthesis and functionalization” are inappropriate as keyword”

Our Answer: “We thank you for this remark. In order to cover the broad range of the content, we provided several keywords. However, we recognized that there may be too many and thus we deleted the proposed and superficial terms.”

Reviewer #1: “In Page 2, the article lacks current research progression the application of photoremovable protecting groups. And what are the advantages of this research compared with other researches?”

Our Answer: “Based on this valuable comment and the exciting research performed in the field of PPGs, we extended our introduction on page 2, line 32 as follows:

An alternative approach employs photoremovable protecting groups (PPGs), which have been reported for various functionalities [36–44]. One of the early PPGs, which found broad application, is based on the 2-nitrobenzyl motif and till the 1980s about six main classes of different PPG classes were described [41]. Until 2013, the spectrum of PPG systems was significantly enhanced as elaborated by the exhaustive review of Klán et al. [36]. Maybe some current research directions worthwhile to point out are as follows: Concurrent implementation of complementary PPGs at solid interfaces, which allow selective deprotection in a lithographic manner [42]; long-wavelength-sensitive PPGs for biological and biomedical in vivo applications [43]; two-photon-sensitive PPGs, which combine the advantages of longer wavelengths for deprotection with high spatial resolution [44]. Generally, PPGs provide the advantage of using light as sole means for cleavage under mild conditions, allowing good spatio-temporal control and obviating the need for chemical reagents in the deprotection reaction [36,40].

The specific advantages of our present research are pointed out on page 2, lines 1 to 26.”

Reviewer #1: “In “Polymer synthesis” section, the authors describe that “the number averaged molar masses (Mn) of the copolymers are congruent with the target molar mass of 15 kg mol-1”. What is the number averaged molar masses (Mn) of the copolymers determined by GPC?”

Our Answer: “We are thankful that you called our attention to a minor confusion regarding the molar masses of our synthesized polymers. We were actually referring to the molar mass at peak maximum (Mp) of our GPC curves, because the broad mass distributions did not allow reliable calculation of Mn. As we discussed later, side reactions of the PPG could have led to several species increasing the signal width significantly. Further, the Mp value decreases with increasing NbMEtOxa content, which is also discussed in detail for the GPC curves in the main text. In the manuscript, we adapted the text on page 5, lines 6-12 accordingly:

The molar mass at peak maximum (Mp) of the copolymers, as determined by GPC (see ESI, Figure S14), matches for the low NbMEtOxa content with the target molar mass of 15 kg mol-1, but decreases with higher comonomer content (Mp for P1%: 16.1 kg mol-1; P2.5%:13.7 kg mol‑1; P5%:8.0 kg mol-1; P10%: 5.6 kg mol-1). For an ideal Gaussian curve, Mncorresponds to Mp, but due to the broader molar mass distributions and the presence of several polymer species Mn is not well defined and thus we provide Mp for the maximum of the GPC trace.

Reviewer #1: “Some figures are not clear enough to be read and the quality should be improved. In Figure S15, I suggest that the GPC curves of the copolymer samples (p1%, p2.5%, p5%, p10%) are placed on a graph to better illustrate the variation trend of molar mass with the increase of NbMEtOxa content.”

Our Answer: “All figures in the manuscript have been exchanged with portable network graphics (PNG) in order to preserve the quality. We understand that the illustration of the GPC curves in one diagram does improve the comparability between the different copolymers and have included such graphs in ESI, Figure S14E and S14F.”

Reviewer #1: “When the copolymer contains a small amount of NbMEtOxa (P1%, P2.5%, P5%), the gel does not form. For the polymers with lower thiol group content, how to improve their gel efficiency?”

Our Answer: “In order to improve gel efficiency for polymers with lower thiol content we currently investigate other types of crosslinkers, initiators and solvents, as indicated in the manuscript on page 9, line 13.

Further studies will be directed to the improvement of the gelation efficiency at lower thiol group content (other types of crosslinkers, initiators, and solvents), extension of the tandem network formation concept to responsive polymer systems, and the exploration of potential applications in materials, biomedical, and surface sciences.

Reviewer #1: “The abbreviation that appears for the first time in this article needs to be written in its full name. For example, “CROP” in Keywords.”

Our Answer: “We have double-checked all abbreviations and changed on page 6, line 16 the abbreviation “MeCN” with the term “acetonitrile”. The keyword CROP was removed (see also first comment and our response above).”

Reviewer #1: “There are some formatting errors and grammatical errors in this paper. For example, in Page 1 Line 24, “(PPG), polymer network” should be corrected as “(PPG); polymer network”. In “Supporting Information” section, “20 µl” should be corrected as “20 µL”. In Page 8 Line 16, “Since in the absence of the photoinitiator no gel formation was observed, spontaneous thiol-Michael addition under these conditions is” needs to be modified. There are other similar mistakes, which aren’t listed here, please revise after checking carefully.”

Our Answer: “Thank you for the helpful advices. We tried to correct every typographical mistake, as it can be followed by the “Track Changes” function in the manuscript file. We have consistently used the lower case letter “l” to indicate the unit liter in agreement with the preferred IUPAC and SI convention.

Furthermore, we changed the sentence on page 8, line 21 as follows:

As no gel formation was observed in the absence of the photoinitiator, spontaneous thiol-Michael addition is excluded under these conditions.

Reviewer #2:

Reviewer #2: “Dear Editor, dear Authors, Niklas Jung et al. submitted a paper on the synthesis and cationic ring-opening copolymerization of an oxazoline monomer 2-{2-[(2-nitrobenzyl)thio]ethyl}-4,5-dihydrooxazole (NbMEtOxa) carrying 2-nitrobenzyl-protected thiol groups at varying ratios with 2-ethyl-2-oxazoline (EtOxa). The authors have fully characterized their copolymers obtained with different NbMEtOxa content (P1%, P2.5%, P5%, P10%) by using 1H NMR spectroscopy, gel permeation chromatography (GPC), and UV/Vis spectroscopy. Furthermore, they have also demonstrated that the resulting polymer show the ability to form a network/gelation under UV light irradiation originated from thiol deprotection via removal of the photolabile 2-nitrobenzyl protecting group and concurrent crosslinking via thiol-ene click reaction coupling, when treated with photoinitators such as 2-hydroxy-2-methylpropiophenone (HMPP) and pentaerythritol tetraacrylate (PETA), a tetrafunctional vinyl crosslinker. For my opinion, the authors have performed an original study that fit with the scope of polymers journal, the manuscript is well written, and results are well supported with experimental evidence. Therefore, I believe that the manuscript can be accepted for publication in Polymers Journal. However, I have very few comments to the authors that should be revised.”

Our Answer: “We also would like to thank reviewer #2 for his constructive comments, which we have all addressed in the revised manuscript as outlined below.”

Reviewer #2: “Page 1, Supporting information “schematic” Please revise the photo of the vial prior to crosslinking as it looks like an empty vial.

Our Answer: “We have enhanced the contrast of the picture and added red arrows to increase the visibility of the liquid in the vial to Figure 3 (manuscript page 8, line 1) and the scheme at the beginning of the supporting information.”

Reviewer #2: “Figure S7, supporting information, how the authors can explain why the integration of the Benzene ring protons is too low (0.77 and 2.78).”

Our Answer: “We attribute this deviation to the standard integration error, in particular for very low signal intensities in the NMR.”

Reviewer 2 Report

Dear Editor, dear Authors, Niklas Jung et al. submitted a paper on the synthesis and cationic ring-opening copolymerization of an oxazoline monomer 2-{2-[(2-nitrobenzyl)thio]ethyl}-4,5-dihydrooxazole (NbMEtOxa) carrying 2-nitrobenzyl-protected thiol groups at varying ratios with 2-ethyl-2-oxazoline (EtOxa). The authors have fully characterized their copolymers obtained with different NbMEtOxa content (P1%, P2.5%, P5%, P10%) by using 1H NMR spectroscopy, gel permeation chromatography (GPC), and UV/Vis spectroscopy. Furthermore, they have also demonstrated that the resulting polymer show the ability to form a network/gelation under UV light irradiation originated from thiol deprotection via removal of the photolabile 2-nitrobenzyl protecting group and concurrent crosslinking via thiol-ene click reaction coupling, when treated with photoinitators such as 2-hydroxy-2-methylpropiophenone (HMPP) and pentaerythritol tetraacrylate (PETA), a tetrafunctional vinyl crosslinker. For my opinion, the authors have performed an original study that fit with the scope of polymers journal, the manuscript is well written, and results are well supported with experimental evidence. Therefore, I believe that the manuscript can be accepted for publication in Polymers Journal. However, I have very few comments to the authors that should be revised.

  • Page 1, Supporting information “schematic” Please revise the photo of the vial prior to crosslinking as it looks like an empty vial
  • Figure S7, Supporting information, how the authors can explain why the integration of the Benzene ring protons is too low (0.77 and 2.78)

Sincerely Yours,

Author Response

(The authors gave the same response as above.)
